# Improving Neural ODE Training with Temporal Adaptive Batch Normalization

**Su Zheng**[1]*, **Zhengqi Gao**[2]*, **Fan-Keng Sun**[2], **Duane S. Boning**[2], **Bei Yu**[1], **Martin Wong**[1]
[1]Department of CSE, CUHK    [2] Department of EECS, MIT

## Abstract

Neural ordinary differential equations (Neural ODEs) is a family of continuous-depth neural networks where the evolution of hidden states is governed by learnable temporal derivatives. We identify a significant limitation in applying traditional Batch Normalization (BN) to Neural ODEs, due to a fundamental mismatch — BN was initially designed for discrete neural networks with no temporal dimension, whereas Neural ODEs operate continuously over time. To bridge this gap, we introduce temporal adaptive Batch Normalization (TA-BN), a novel technique that acts as the continuous-time analog to traditional BN. Our empirical findings reveal that TA-BN enables the stacking of more layers within Neural ODEs, enhancing their performance. Moreover, when confined to a model architecture consisting of a single Neural ODE followed by a linear layer, TA-BN achieves 91.1% test accuracy on CIFAR-10 with 2.2 million parameters, making it the first `unmixed` Neural ODE architecture to approach MobileNetV2-level parameter efficiency. Extensive numerical experiments on image classification and physical system modeling substantiate the superiority of TA-BN compared to baseline methods.

## 1   Introduction

Originally derived as the continuous limit of a residual connection [4], neural ordinary differential equations (Neural ODEs) [4, 7, 37, 45, 8, 42, 30, 18, 32, 19, 31] is a family of continuous-depth neural networks where the evolution of hidden states is governed by learnable temporal derivatives. These models exhibit several intriguing features, such as the capability of temporal reversibility, which enables generative modeling [4, 15].

Previous studies on Neural ODEs parameterize the learnable temporal derivatives using a shallow neural network with a limited number of parameters [7, 30]. Without special treatment, merely stacking additional layers in the temporal derivatives does not necessarily enhance Neural ODE performance. Furthermore, deeper networks might increase the stiffness of the ODE system, leading to challenges with the ODE solver, such as excessively small step sizes or even failures due to infinite state values, as shown in Figure 1.

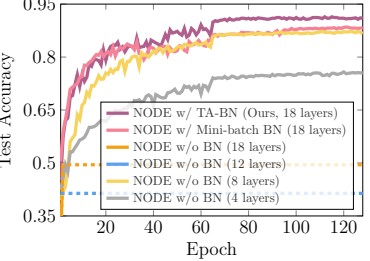

Figure 1: Test accuracies are depicted over the training epochs on CIFAR-10, utilizing Neural ODEs with different numbers of convolution layers as the backbones for learnable derivatives. Dashed horizontal lines denote instances of training failures.

In efforts to deepen the model and address instabilities, one might instinctively consider Batch Normalization (BN) [5, 29, 16, 39, 2] as a remedy to stabilize the intermediate state values of Neural ODEs [13, 35], because BN was originally proposed to accelerate

---

*The first two authors contribute equally. Correspond to Bei Yu at byu@cse.cuhk.edu.hk.

38th Conference on Neural Information Processing Systems (NeurIPS 2024).

the training of (discrete) neural networks and consolidate them against internal covariate shift [29]. However, preliminary works surprisingly found that the introduction of BN into Neural ODEs can even degrade performance, an effect for which the underlying causes remain unclear [13, 35].

In this paper, we demystify the aforementioned phenomenon and demonstrate that directly applying traditional BN to Neural ODEs is fundamentally flawed. The primary complication arises because the forward pass of Neural ODEs employs an adaptive step size solver that discretizes time variably. Consequently, it is not assured that two forward passes will coincide with the same discretized time grids. This variability precludes the possibility of retrieving *population statistics* [16] at test time. Additionally, relying on *mini-batch statistics* for BN renders the outputs of Neural ODEs non-deterministic and vulnerable to outliers and small batch size.

Motivated by these observations, we introduce temporal adaptive BN (TA-BN), a novel technique designed as the continuous-time counterpart to traditional BN, tailored specifically for Neural ODEs. Our empirical findings demonstrate that TA-BN facilitates the up-scaling of Neural ODE model sizes without encountering performance saturation (See Figure 1). Moreover, when confined to a model architecture consisting of a single Neural ODE followed by a linear layer, TA-BN achieves 91.1% test accuracy on CIFAR-10 with 2.2 million parameters, making it the first `unmixed` Neural ODE architecture to approach MobileNetV2-level parameter efficiency. Extensive numerical experiments on image classification and physical system modeling substantiate the superiority of TA-BN compared to baseline methods.

## 2 Preliminary

**Neural ODEs**    Neural ODEs [4, 7, 37, 45, 8, 42, 30, 18, 32, 19, 31] forge a significant linkage between differential equations and neural networks. They model the continuous dynamics of hidden states with a learnable ODE system:

$$\frac{d\mathbf{h}(t)}{dt} = \mathbf{f}_{\boldsymbol{\theta}}(\mathbf{h}(t), t) \tag{1}$$

where $\mathbf{f}_{\boldsymbol{\theta}}(\cdot, \cdot) \in \mathbb{R}^D \times \mathbb{R} \to \mathbb{R}^D$ is a neural network parameterized by learnable parameter $\boldsymbol{\theta}$ controlling the temporal evolution, and $\mathbf{h}(t) \in \mathbb{R}^D$ is the ODE state variable at time $t$. The practical usage of Neural ODEs is by solving an initial value problem: given the initial state $\mathbf{h}(0)$ as input, the state $\mathbf{h}(T)$ at a later time $T$ is computed using an ODE solver and returned as the output. The gradients required for training are computed via the adjoint method [4, 31, 18] reverse in time.

Subsequent works have expanded the capabilities and scope of Neural ODEs by introducing modifications, such as improving expressivity through state augmentation [7], coupling the evolution of both weights and activations [43], accelerating convergence via semi-norm techniques [18], extending to graph neural networks [37] and second-order ODEs [34], adapting to irregular time-series data [19, 32], and drawing connections to diffusion models [15, 28]. Of particular interest, two preliminary studies [13, 35] have indicated that applying BN within Neural ODEs may compromise model performance. Our work aims to unravel this phenomenon and propose a version of BN tailored to operate effectively with Neural ODEs.

**Batch Normalization**    BN [5, 29, 16, 39, 2, 21] performs a re-centering and a re-scaling operation on the given input by subtracting the mean and dividing by the standard deviation:

$$\text{BN}(x_i) = \text{BN}_{\gamma, \alpha}(x_i) = \frac{x_i - \mu}{\sqrt{\sigma^2 + \epsilon}} \gamma + \alpha \tag{2}$$

where $\epsilon \in \mathbb{R}^+$ ensuring division validity, $\{\gamma \in \mathbb{R}, \alpha \in \mathbb{R}\}$ are learnable parameters, and $x_i \in \mathbb{R}$ represents the $i$-th data in one batch of size $B$. During training, the mini-batch statistics [2] $\mu = 1/B \sum_{i=1}^{B} x_i$ and $\sigma^2 = 1/B \sum_{i=1}^{B} (x_i - \mu)^2$ are used for efficiency. In contrast, at test time, BN uses population statistics aggregated across the entire training dataset to ensure deterministic outputs [16]. BN can be extended to multiple dimensions/channels by processing each one separately.

BN has been widely applied in training deep neural networks, as it can stabilize the training process and accelerate convergence. Despite its strong empirical performance, the underlying mechanisms

---

[2]Throughout our paper, statistics refer to mean and standard deviation (std) or equivalently variance.

of BN have been subject to various interpretations, such as reducing internal covariate shift [16], smoothing the optimization landscape [39], and decoupling the learning of length and direction [21]. Originally, BN [16] was developed for neural networks with hidden units that do not explicitly depend on time. However, recent advancements have extended BN to accommodate time-dependent models, including recurrent neural networks (RNNs)[5, 23] and spiking neural networks (SNNs) [20, 44, 6, 17]. Unlike these models, Neural ODEs exploit adaptive time discretization, necessitating a specialized modification of BN for its application.

## 3 Traditional Batch Normalization in Neural ODEs

Let us consider a simplified one-dimensional Neural ODE when incorporating BN as example:

$$\frac{dh}{dt} = \text{BN}_{\gamma,\alpha}(wh + b) \tag{3}$$

where $\boldsymbol{\theta} = \{\gamma \in \mathbb{R}, \alpha \in \mathbb{R}, w \in \mathbb{R}, b \in \mathbb{R}\}$ are the learnable parameters. Given a batch of input $\{h_i(0)\}_{i=1}^B$ (i.e., initial values of the ODE), the forward process employs an ODE solver to discretize Eq. (3) over time, visiting $N$ sequential points $\mathcal{T} = (t_0 = 0, t_1, t_2, \cdots, t_N = T)$ and computing the outputs $\{h_i(T)\}_{i=1}^B$. An adaptive step size solver (e.g., Runge-Kutta method) is usually preferred because of its higher efficiency compared to a fixed step size solver (e.g., Forward Euler method). Thus, the differences between two consecutive time steps might not be identical (e.g., $t_2 - t_1 \neq t_1 - t_0$), and the value of $N$ also depends on the batch of data. The temporal discretization indicates that we need to invoke BN at every $t \in \mathcal{T}$ for every sample:

$$\text{BN}(x_{i,j}) = \text{BN}_{\gamma,\alpha}(x_{i,j}) = \frac{x_{i,j} - \mu_j}{\sqrt{\sigma_j^2 + \epsilon}}\gamma + \alpha, \quad \text{where } x_{i,j} = w \cdot h_i(t_j) + b \tag{4}$$

for every $j = 1, 2, \cdots, N$ and $i = 1, 2, \cdots, B$, where $\mu_j$ and $\sigma_j^2$ are statistics associated with $t_j$.

**During training** To perform the normalization shown in Eq. (4), BN applies mini-batch mean $\mu_j = 1/B \sum_{i=1}^B x_{i,j}$ and variance $\sigma_j^2 = 1/B \sum_{i=1}^B (x_{i,j} - \mu_j)^2$. Although the training can proceed under normal conditions, the statistics cannot be successfully accumulated across batches. For instance, the time grid $\mathcal{T}$ utilized for the first batch might differ significantly from the grid $\mathcal{T}'$ used for a subsequent batch. Consequently, specific time points $t_j' \in \mathcal{T}'$ might not coincide with any time points in $\mathcal{T}$, and vice versa. This might result in an impractical collection of infinite statistics $\mu_j$ and $\sigma_j^2$. Moreover, except those associated with $t = 0$ and $t = T$, most stored statistics will be calculated and updated based on a limited number of batches.

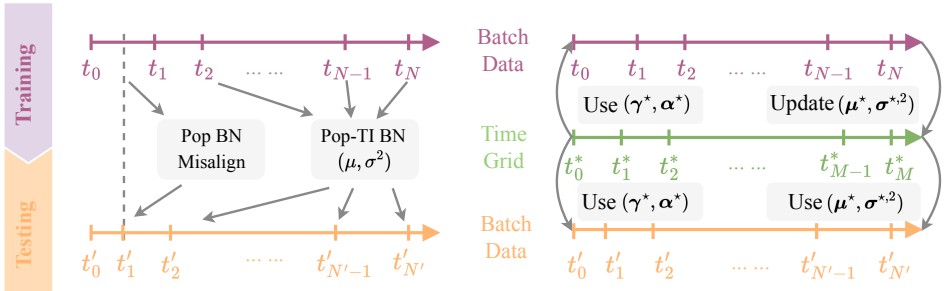

Figure 2: Left: The failure of Pop BN in Neural ODEs stems from the misalignment of discretized time grids. Pop-TI BN aggregates all running mini-batch statistics into a single pair of $(\mu, \sigma^2)$, implicitly assuming time-independent population statistics. Right: Our proposed TA-BN automatically conducts temporal interpolation to accumulate statistics and update parameters during training and testing.

**During inference** Conventionally, BN should perform the normalization shown in Eq. (4) using population statistics during inference. However, as shown in the left part of Figure 2, the population statistics associated with the time point $t_j' \in \mathcal{T}'$, required by the temporal discretization during inference, might not be available if the time value $t_j'$ is never encountered during training. Moreover, even if the population statistics exist, they are likely to be inaccurate as discussed above.

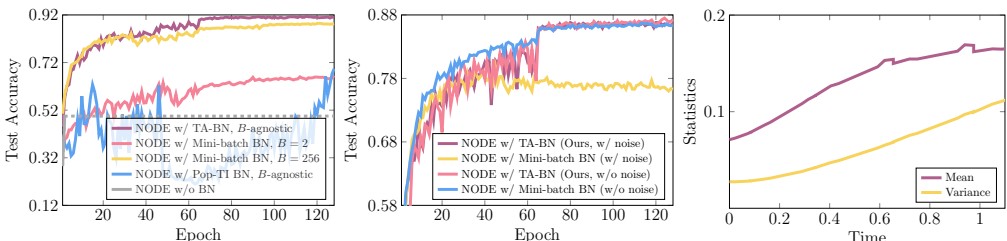

Figure 3: Left: We train a Neural ODE with a U-Net backbone as the learnable derivatives on CIFAR-10. Mini-batch BN shows degraded accuracies with a batch size of 2, while TA-BN can maintain high accuracies under varying batch sizes, because it uses the estimated population statistics during testing. Pop-TI BN aggregates running statistics encountered at any time points into a single pair of $\mu$ and $\sigma^2$. This approach assumes time-independent statistics, leading to erroneous predictions and erratic test loss curves. Middle: When noisy data exist on average in one out of every test batch, Mini-batch BN's performance deteriorates, because the noise affects the mini-batch statistics. The backbone for learnable derivatives in this experiment consists of 6 convolution layers. Right: We plot the output statistics from the first layer of U-Net over time; they are time-dependent.

**Summary**  A seemingly straightforward remedy could be to consistently use mini-batch statistics during both training and inference, thus avoiding the need for accumulating population statistics. However, this approach has a critical limitation: the output $h_i(T)$ becomes highly dependent on their respective batch's specific characteristics. This will make the outputs inaccurate when batch size is very small (e.g., $B = 2$) or the occurrence of outliers in batch, and even fail in single-run scenarios ($B = 1$) because running standard deviation cannot be calculated. As shown in the left of Figure 3, when the batch size is reduced to 2, mini-batch BN displays degraded accuracies. Furthermore, when noisy data exist, mini-batch BN performance deteriorates, as shown in the middle of Figure 3.

For clarity, we will henceforth refer to the use of population statistics at test time as Pop BN, and the use of mini-batch statistics as Mini-batch BN. Notably, both methods utilize mini-batch statistics during training. It is important to emphasize that in practice, when implementing BN in Neural ODEs with popular deep learning frameworks (e.g., PyTorch with TorchDiffeq [4]), actually a variant of Pop BN is used, which we will refer to as Pop-TI BN. Pop-TI BN accumulates statistics at all encountered time points during training into a single pair of $\mu$ and $\sigma^2$, which implicitly assumes that all time points share the same statistics. This aggregation approach can result in outputs during test time without errors concerning missing population statistics which we would anticipate as in Pop BN. Unfortunately, these predictions are meaningless and often lead to poor performance, as illustrated in the left of Figure 3. This issue highlights the challenges identified in recent studies [13, 35]. [3] In a nutshell, our observations can be succinctly summarized as follows:

- Pop BN fails when Neural ODEs employ an adaptive step size solver.
- Pop-TI BN implicitly assumes time-independent statistics, leading to erroneous outputs.
- Mini-batch BN fails when Neural ODEs operate with small batch sizes or outliers exist.

## 4   Temporal Adaptive Batch Normalization

The fundamental reason why Pop-TI BN and Mini-batch BN fail in Neural ODEs is attributed to the added temporal dimension, where the distributions at different time points vary, effectively constituting a stochastic process as depicted in the right of Figure 3. This observation suggests that BN should be defined to be time-dependent in Neural ODEs. Ideally, we would want BN to store population statistics at every $t \in [0, T]$, with corresponding learnable parameters $\gamma(t)$ and $\beta(t)$ defined at every $t$. However, implementing such a model is impractical, because not every $t$ will be encountered in training, and the update of $\gamma(t)$, $\beta(t)$, and the population statistics will be naturally sparse on $t$, resulting in inefficiency.

---

[3]We examined several open-source codes provided by previous works. They align with this implementation and exhibit similar erratic behavior to the Pop-TI BN's test loss curve in the left of Figure 3.

To address the aforementioned problem, we propose an intuitive method termed temporal adaptive BN (TA-BN) for usage in Neural ODEs. Let us illustrate it with the example shown in Eq. (3). To begin with, we evenly divide the time span $[0, T]$ defining $(M + 1)$ time grids $\mathcal{T}^\star = (t_0^\star = 0, t_1^\star = \frac{T}{M}, t_2^\star = \frac{2T}{M}, \cdots, t_M^\star = T)$. We associate the time grid $t_m^\star$ with population mean $\mu_m^\star$ and population variance $\sigma_m^{\star,2}$, as well as learnable parameters $\gamma_m^\star$ and $\alpha_m^\star$ for every $m = 0, 1, 2, \cdots, M$. For later simplicity, we denote $\boldsymbol{\gamma}^\star = [\gamma_0^\star, \gamma_1^\star, \cdots, \gamma_M^\star]^T$ and $\boldsymbol{\alpha}^\star = [\alpha_0^\star, \alpha_1^\star, \cdots, \alpha_M^\star]^T$. Similar notations apply to $\boldsymbol{\mu}^\star$ and $\boldsymbol{\sigma}^\star$.

**During training** As shown in the right of Figure 2, given a batch of data $\{h_i(0)\}_{i=1}^B$, the forward pass of Neural ODE might discretize the time as $\mathcal{T} = (t_0, t_1, \cdots, t_N)$ which differs from the grids $\mathcal{T}^\star$ of TA-BN. However, TA-BN can calculate the temporal derivative for the $i$-th data at $t_j$:

$$\text{TABN}_{\boldsymbol{\gamma}^\star, \boldsymbol{\alpha}^\star}(x_{i,j}) = \frac{x_{i,j} - \mu_j}{\sqrt{\sigma_j^2 + \epsilon}}\gamma_j + \alpha_j \quad \text{where } x_{i,j} = w \cdot h_i(t_j) + b \tag{5}$$

where $(\mu_j, \sigma_j^2)$ are still the mini-batch mean and variance based on $\{x_{i,j}\}_{i=1}^B$. Here $\gamma_j$ and $\alpha_j$ are interpolated based on $\boldsymbol{\gamma}^\star$ and $\boldsymbol{\alpha}^\star$, respectively:

$$\gamma_j = G(t_j, \boldsymbol{\gamma}^\star, \mathcal{T}^\star), \quad \alpha_j = G(t_j, \boldsymbol{\alpha}^\star, \mathcal{T}^\star) \tag{6}$$

where $G(t, \mathbf{a}, \mathcal{T})$ is a function to interpolate the value $a(t)$ given an array of values $\mathbf{a} \in \mathbb{R}^{M+1}$ and their corresponding $(M + 1)$ time points $\mathcal{T}$. There are many choices for $G(\cdot, \cdot, \cdot)$, such as linear, cubic spline, and kernel smoothing. We empirically observe that linear interpolation based on two nearest neighbors suffices for our experiments, and we advocate it due to its implementation simplicity:

$$G(t, \mathbf{a}, \mathcal{T}) = \frac{t_{l+1} - t}{t_{l+1} - t_l}a_l + \frac{t - t_l}{t_{l+1} - t_l}a_{l+1} \tag{7}$$

where $l$ is the index which makes $t_l$ to be the largest value smaller than $t$ in $\mathcal{T}$. Using a linear $G(\cdot, \cdot, \cdot)$ implies that we approximate the underlying $a(t)$ in a piece-wise linear manner, and as long as the number of time grids is sufficiently large, $G(t, \mathbf{a}, \mathcal{T})$ can approximate any continuous $a(t)$ arbitrarily well, i.e., $|G(t, \mathbf{a}, \mathcal{T}) - a(t)| \to 0$ when $M \to \infty$. In practice, we set $M$ close to the number of ODE-discretized time grids. Please refer to Section 5 for ablations on the choice of $G(\cdot, \cdot, \cdot)$.

During training, we also need to accumulate the encountered running mini-batch statistics $(\mu_j, \sigma_j^2)$ to the population statistics $(\mu_m^\star, \sigma_m^{\star,2})$, so that later they can be used for inference. As shown in the right part of Figure 2, this requires the interpolation to be performed in the opposite direction:

$$\begin{aligned} \mu_m^\star &\leftarrow (1 - \eta) \cdot \mu_m^\star + \eta \cdot G(t_m^\star, \boldsymbol{\mu}, \mathcal{T}), &\boldsymbol{\mu} &= [\mu_1, \mu_2, \cdots, \mu_N]^T \\ \sigma_m^{\star,2} &\leftarrow (1 - \eta) \cdot \sigma_m^{\star,2} + \eta \cdot G(t_m^\star, \boldsymbol{\sigma^2}, \mathcal{T}), &\boldsymbol{\sigma^2} &= [\sigma_1^2, \sigma_2^2, \cdots, \sigma_N^2]^T \end{aligned} \tag{8}$$

where $\eta \in [0, 1]$ is a momentum constant to perform moving averaging to update the population statistics based on current estimate of $\mu_m^\star$ and $\sigma_m^{\star,2}$ using the new observed values $G(t_m^\star, \boldsymbol{\mu}, \mathcal{T})$ and $G(t_m^\star, \boldsymbol{\sigma^2}, \mathcal{T})$, respectively. Note that the training does not involve the gradients with respect to $t$.

**During inference** After the training phase, TA-BN will have well-trained parameters $\boldsymbol{\gamma}^\star$ and $\boldsymbol{\alpha}^\star$, and accurately record population statistics $\boldsymbol{\mu}^\star$ and $\boldsymbol{\sigma}^\star$. During inference, when provided with a batch of data and denoting the discretized time grids as $\mathcal{T}'$, as depicted in the right part of Figure 2, we again proceed with the interpolation step:

$$\begin{aligned} \text{Param.} \quad & \gamma_j' = G(t_j', \boldsymbol{\gamma}^\star, \mathcal{T}^\star), & \alpha_j' = G(t_j', \boldsymbol{\alpha}^\star, \mathcal{T}^\star) \\ \text{Pop. Stat.} \quad & \mu_j' = G(t_j', \boldsymbol{\mu}^\star, \mathcal{T}^\star), & \sigma_j'^{,2} = G(t_j', \boldsymbol{\sigma}^{\star,2}, \mathcal{T}^\star) \end{aligned} \tag{9}$$

This interpolation differs from the training phase in that now the interpolation directions are the same for parameters and the population statistics, because now we utilize them for the new time grids $\mathcal{T}'$.

**Implementations and summary** In practice, we observe that the interpolations during training outlined in Eq. (6)-(8) have opposite directions, potentially leading to increased computational overhead. To elaborate, when encountering a time point $t_j$ discretized by the ODE solver during training, the interpolation for obtaining $\gamma_j$ and $\alpha_j$ in Eq. (6) using Eq. (7) can be immediately executed by identifying the index $l$ such that $t_l^\star$ is the largest value smaller than $t_j$. However, the interpolation

in Eq. (8) cannot be carried out at this time. It becomes feasible only after the ODE solver completes its execution and we gather all $t_j$'s into $\mathcal{T}$. This sequential nature degrades computational efficiency. To mitigate this issue, we have slightly adjusted the interpolation for population statistics during training so that it can be performed concurrently with the parameter interpolation at time $t_j$. The key steps of our proposed TA-BN are summarized in Algorithm 1. It will be called as a subroutine for every discretized time point $t_j \in \mathcal{T}$ required by the ODE solver.

Algorithm 1 outlines the case involving a single one-dimensional neuron and one TA-BN layer within the Neural ODE. In more general cases, we may have $Q$ TA-BN layers within a Neural ODE, and each neuron may possess $D$ dimensions. In such cases, we can perform the normalization in a dimensional-wise manner. Thus, we must maintain $2(M+1)Q$ population statistics and $2(M+1)Q$ learnable parameters, each with $D$ dimensions. In contrast, a traditional feedforward neural network with $Q$ BNs has $2Q$ population statistics and $2Q$ learnable parameters, each with $D$ dimensions. Note that for image inputs, we perform channel-wise normalization, so $D$ will be the number of channels.

Finally, since our linear interpolation in Algorithm 1 relies on two nearest neighbors, it is prudent to ensure that the number of TA-BN time grids $(M+1)$ are roughly at the same level as the number of ODE-discretized time grids. Denser grids would be unnecessary as those not adjacent to any $t_j$ would remain unused. Alternatively, as our numerical results will demonstrate, employing other interpolation methods that require all time grids may lead to increased computational overhead without significant improvements. Additionally, we employ L2 regularization $\|\boldsymbol{\gamma}^\star - 1\|^2$ and $\|\boldsymbol{\alpha}^\star\|^2$ to avoid significant discrepancies among the elements in $\boldsymbol{\gamma}^\star$ and $\boldsymbol{\alpha}$, making the training more stable.

---

**Algorithm 1** The forward pass of a TA-BN layer at time $t_j$

---

**Input:** Batched input $\mathbf{x} = \{x_{i,j}\}_{i=1}^B$ at time $t_j$
1: Get $t_l^\star$ and $t_{l+1}^\star$ such that $t_l^\star$ is the largest value smaller than $t_j$ in $\mathcal{T}^*$ ;
2: $\omega_1 = \frac{t_{l+1}^\star - t_j}{t_{l+1}^\star - t_l^\star}, \omega_2 = \frac{t_j^\star - t_l}{t_{l+1}^\star - t_l^\star}$      $\triangleright$ Compute the weights for linear interpolation;
3: **if** the model is in the `training` mode **then**
4:    $\mu_j = \text{mean}(\mathbf{x}), \quad \sigma_j^2 = \text{var}(\mathbf{x})$ ;                  $\triangleright$ Compute the mini-batch statistics;
5:    $\gamma_j = \omega_1 \gamma_l^\star + \omega_2 \gamma_{l+1}^\star, \quad \alpha_j = \omega_1 \alpha_l^\star + \omega_2 \alpha_{l+1}^\star$ ;
6:    $\mu_l^\star \leftarrow (1 - \eta \cdot \omega_1)\mu_l^\star + \eta \cdot \omega_1 \mu_j, \quad \mu_{l+1}^\star \leftarrow (1 - \eta \cdot \omega_2)\mu_{l+1}^\star + \eta \cdot \omega_2 \mu_j$ ;
7:    $\sigma_l^{\star,2} \leftarrow (1 - \eta \cdot \omega_1)\sigma_l^{\star,2} + \eta \cdot \omega_1 \sigma_j^2, \quad \sigma_{l+1}^{\star,2} \leftarrow (1 - \eta \cdot \omega_2)\sigma_{l+1}^{\star,2} + \eta \cdot \omega_2 \sigma_j^2$ ;
8: **else if** the model is in the `evaluation` mode **then**
9:    $\mu_j = \omega_1 \mu_l^\star + \omega_2 \mu_{l+1}^\star, \quad \sigma_j^2 = \omega_1 \sigma_l^{\star,2} + \omega_2 \sigma_{l+1}^{\star,2}$ ;
10:    $\gamma_j = \omega_1 \gamma_l^\star + \omega_2 \gamma_{l+1}^\star, \quad \alpha_j = \omega_1 \alpha_l^\star + \omega_2 \alpha_{l+1}^\star$ ;
11: **end if**
12: **return** $\frac{\mathbf{x} - \mu_j}{\sqrt{\sigma_j^2 + \epsilon}}\gamma_j + \alpha_j$ ;

---

## 5 Numerical Results

Various network architecture designs have been explored in existing Neural ODE studies. For instance, the original study [4] integrates a feature extractor (e.g., CNN-based), a Neural ODE module, and an MLP in sequence for classification. Another study structures neural networks with alternating Neural ODE modules and convolutional/linear layers [11]. While these solutions represent viable approaches for optimizing performance on given ML tasks through the synergy of conventional (discrete) neural networks and Neural ODEs, they become inappropriate for investigating the effect of specific architectural modifications (i.e., TA-BN) on Neural ODEs. This is because the Neural ODE module can be entirely bypassed if its trained weights approach zero in extreme cases, thereby overshadowing the intended modification. Hence, our model comprises **a Neural ODE module followed by a single linear layer** as advocated by [7], which we refer to as the `unmixed` Neural ODE architecture. Our code is developed based on PyTorch [36], TorchDiffeq [3], and a customized TA-BN layer. All experiments are run on a Linux server with RTX 3090 GPUs.

### 5.1 Image Classification

Being consistent in experimental scales of previous Neural ODE studies, we conduct image classification across datasets including MNIST [26], SVHN [33], CIFAR-10, CIFAR-100 [22], and

Tiny-ImageNet [24]. We employ the dopri5 solver with a tolerance of $10^{-3}$ for ODE solving and adopt the AdamW optimizer [27] with a learning rate of $10^{-3}$ to train the neural networks for 128 epochs. The training batch size is 256. We set $M = 100$ for TA-BN.

Table 1 compares the test top-1 accuracy and number of parameters of various Neural ODEs. TA-BN outperforms Mini-batch BN and Neural ODE w/o BN on SVHN, CIFAR-10, CIFAR-100, and Tiny-ImageNet. Moreover, TA-BN achieves superior accuracies over Aug-NODE [7] and STEER [12]. On MNIST and CIFAR-10, TA-BN has fewer parameters than Aug-NODE and STEER, which indicates that TA-BN can also help training in the small Neural ODE regime. We also visualize Neural ODEs' performance changes as the number of parameters varies in Figure 4, by including four additional baselines IL-NODE, 2nd-Ord [30], HBNODE, and GHBNODE [41]. For reference, the popular convolutional-based MobileNetV2 [38], known for its parameter efficiency, achieves approximately 94% accuracy with about 2M parameters. Our TA-BN assisted Neural ODE is the first to approach this level of performance using the unmixed architecture, while most previous Neural ODE literature either falls short of this performance or/and does not follow this unmixed architecture.

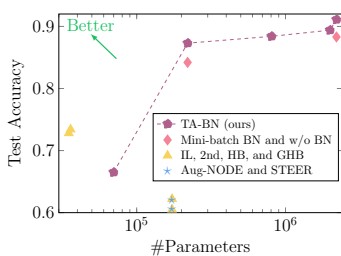

Figure 4: Comparison between different Neural ODEs on CIFAR-10. The baselines marked by yellow triangles do not adhere to the unmixed structure and are not strictly comparable to ours. It is unknown whether increasing the number of parameters inside their ODEs can lead to better accuracy.

Table 1: Comparison of test accuracies and number of parameters between different Neural ODEs[†].

| Model | MNIST | | CIFAR10 | | SVHN | | CIFAR100 | | Tiny-Imagenet | |
|---|---|---|---|---|---|---|---|---|---|---|
| | Accuracy | #Params | Accuracy | #Params | Accuracy | #Params | Accuracy | #Params | Accuracy | #Params |
| Aug-NODE [7] | 0.982 | 84k | 0.606 | 172k | 0.835 | 172k | N/A | N/A | N/A | 366k |
| STEER [12] | 0.986 | 84k | 0.621 | 172k | 0.841 | 172k | N/A | N/A | N/A | N/A |
| w/o BN | **0.989±0.001** | 37k | 0.517±0.049 | 2.2M | 0.096±0.025 | 2.2M | 0.246±0.084 | 2.2M | - | 2.2M |
| w/ Pop-TI BN | 0.973±0.011 | 37k | 0.548±0.087 | 2.2M | 0.241±0.123 | 2.2M | 0.251±0.112 | 2.2M | 0.044±0.007 | 2.2M |
| w/ Mini-batch BN | 0.962±0.013 | 37k | 0.822±0.095 | 2.2M | 0.906±0.031 | 2.2M | 0.492±0.176 | 2.2M | 0.200±0.006 | 2.2M |
| w/ TA-BN | 0.988±0.001 | 37k | 0.748±0.059 | 70k | 0.953±0.002 | 220k | 0.576±0.016 | 220k | 0.436±0.013 | 220k |
| (ours) | 0.988±0.001 | 220k | **0.910±0.010** | 2.2M | **0.958±0.004** | 2.2M | **0.664±0.025** | 2.2M | **0.512±0.008** | 2.2M |

[†] 'N/A' indicates values are not available in the original literature. '-' indicates training failure at the first epoch. Results are reported with a test batch size of 256. The error bars are shown using the format mean±std.

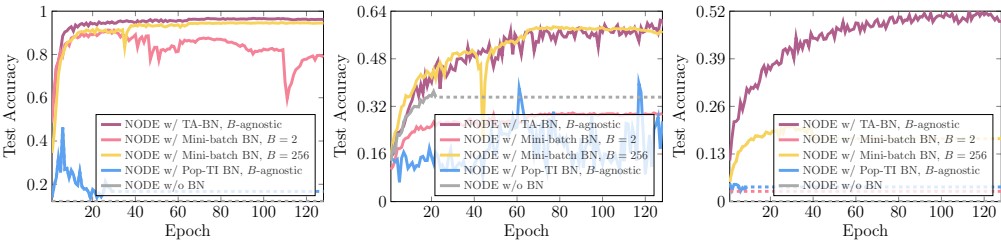

Figure 5: Comparison between Neural ODEs w/ TA-BN, w/ Mini-batch BN, w/ Pop-TI BN, and w/o BN, using an 18-layer U-Net backbone for learnable derivatives. The total number of parameters including the outside linear layer is 2.2M. The datasets are SVHN (left), CIFAR-100 (middle), and Tiny-ImageNet (right). The results on CIFAR-10 have been shown in the left part of Figure 3.

The left part of Figure 3 in Section 3 monitors the test accuracy over training epochs on CIFAR-10. It indicates that training a large Neural ODE without special processes might fail due to numerical instability. Pop-TI BN performs poorly because of the time-independent statistics assumption. While Mini-batch BN achieves satisfactory accuracy, our TA-BN outperforms it without encountering the issues associated with Mini-batch BN, such as batch-dependent outcomes and vulnerability to outliers. Similarly, the left and middle parts of Figure 5 display test accuracy versus training epochs on the SVHN and CIFAR-100 datasets. TA-BN consistently shows better and more stable accuracy than Mini-batch BN on these datasets. On the Tiny-ImageNet dataset (right part of Figure 5), TA-BN demonstrates superior accuracy and faster convergence.

**Neural ODE Up-scaling Enabled by TA-BN** As shown in the left part of Figure 6, Neural ODEs with a limited number of layers perform normally with acceptable accuracy; however, when the layer count exceeds 10, training fails due to numerical instability. In contrast, the incorporation of TA-BN enables deeper layers within Neural ODE as the learnable derivatives, scaling up the model size and enhancing accuracy, as exemplified by the middle and right sections of Figure 6. Please see Appendix A.1 for architecture details.

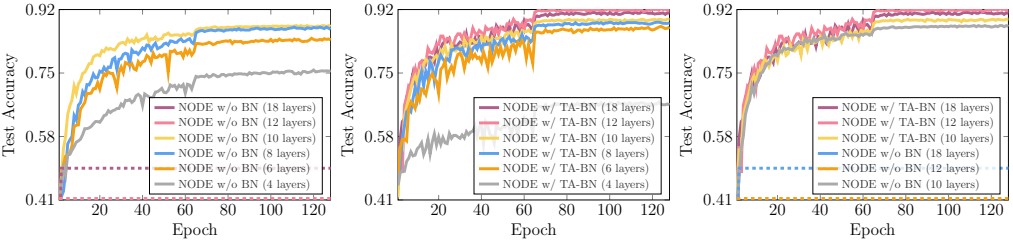

Figure 6: CIFAR-10 accuracies with increasing sizes of the backbones for learnable derivatives. These figures illustrate the scaling up of Neural ODEs without BN (left) and Neural ODEs with TA-BN (middle). We also compare the accuracies of these two settings in one figure (right).

**Ablations on TA-BN** We integrate TA-BN into Aug-NODE [7], HBNODE [41], and SON-ODE [34]. Table 2 reports model accuracies w/ and w/o TA-BN. It indicates that our method can also work in compatible with existing Neural ODE variants and boost their performances.

Table 2: Comparison between Neural ODE variations with and without TA-BN on CIFAR-10$^\dagger$.

| Accuracy | Aug-NODE | HBNODE | SONODE |
|---|---|---|---|
| w/o TA-BN | 0.848 | 0.851 | 0.302 |
| w/ TA-BN | **0.862** | **0.867** | **0.853** |

$^\dagger$ We use 6 convolution layers as the backbone for learnable derivatives.

To show the impact of different interpolation functions $G$, we test the following interpolation methods: (1) Linear interpolation demonstrated by Eq. (7). (2) Cubic interpolation: We use cubic spline interpolation [1]. (3) Kernel smoothing: Given time $t$, we calculate $G(t, \mathbf{a}, \mathcal{T})$ by the weighted sum of elements in $\mathbf{a}$. The weight coefficient for $t_i$ is $K(t, t_i) = \exp(-0.5b^{-2}(t - t_i)^2)$ with $b = 0.1T$. (4) Gaussian process: We fit a Gaussian process [10] to perform interpolation in each forward pass. As illustrated by Figure 7, in addition to linear interpolation, the cubic interpolation and kernel smoothing methods also achieve good performance. However, these methods can be much slower than linear interpolation due to more complicated interpolation mechanisms. With a larger network architecture, we observe that these methods suffer from unaffordable running time and instable performance. Thus, we adopt linear interpolation as our default setting. Please see Appendix B for further discussion. Future work can focus on improving the interpolation strategy for TA-BN.

## 5.2 Physical System Modeling

We first evaluate TA-BN on physical dynamical system modeling tasks using the Walker2d-v2 [25] and HalfCheetah-v2 [14] datasets. These datasets consist of trajectories of 3D robot systems generated by the Mujoco physics engine [40]. Following [42], we perform temporal autoregressive prediction and use the treatments from [42] to avoid collisions. We employ TA-BN in conjunction with a 12-layer MLP serving as the backbone for learnable derivatives. TA-BN with $M = 100$ is applied to the first 5 layers, as is done with Mini-batch BN and Pop-TI BN. We utilize the dopri5 solver with its default settings for solving ODEs and employ the RMSprop optimizer with a learning rate of $10^{-3}$ for training the neural networks over 100 epochs. The batch size is 64 for Walker2d-v2 and 32 for HalfCheetah-v2. Figure 8 presents the mean absolute error (MAE) along the training epochs on these datasets. Compared with Neural ODEs w/o BN, w/ Mini-batch BN, and w/ Pop-TI BN, using TA-BN can achieve lower errors and faster convergence. Moreover, Table 3 summarizes the MAE results and compares them with the results in [42].

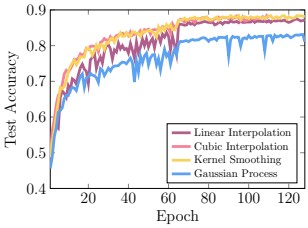

Figure 7: Ablations on interpolation method using CIFAR-10 with 6 convolution layers as the backbone for learnable derivatives. Linear interpolation has a gap of only 0.01 compared to the best result. However, other methods have degraded accuracies and unaffordable runtime when we up-scale the model.

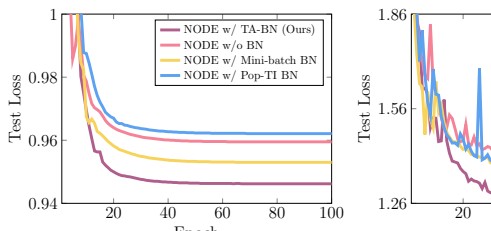
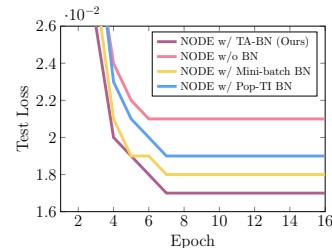

Figure 8: Comparison of Neural ODEs with TA-BN and baselines on Walker2d-v2 (left) and HalfCheetah-v2 (right). The Neural ODE backbone for learnable derivatives is an MLP with 12 layers. Please see Appendix A.2 for architecture details.

Figure 9: Comparison of different Neural ODEs on Charge Pump circuit modeling. The backbone for learnable derivatives is an 8-layer MLP.

Additionally, we investigate the effectiveness of TA-BN by modeling the temporal electrical current of a Charge Pump (CP) circuit [9]. The CP is an analog integrated circuit composed of MOS transistors and is commonly utilized in commercial electrical products. To simulate the semiconductor manufacturing impact on the CP, we generate 200 circuit instances based on the process design kit (PDK) file. Concretely, we simulate the CP's behavior from 0 seconds to 200 nanoseconds using a time grid of 2 picoseconds and record the electrical currents at the CP output every 200 picoseconds.

Table 3: MAE comparison between different Neural ODEs on Walker2d-v2 and HalfCheetah-v2.

| Dataset | Walker2d-v2 | HalfCheetah-v2 |
|---|---|---|
| Baseline [42] | 1.02 | 1.46 |
| w/o BN | 0.959 | 1.40 |
| w/ Mini-batch BN | 0.953 | 1.38 |
| w/ Pop-TI BN | 0.962 | 1.37 |
| w/ TA-BN (ours) | **0.946** | **1.28** |

In this experiment, we utilize 16 circuit features (representing the widths of the MOS transistors) and the electrical currents at the present 20 time points (with each point comprising 2 values) as the Neural ODE input, resulting in 56 input features in total. Our objective is to train the Neural ODE to predict the electrical currents at the subsequent 20 time points, yielding 40 outputs in total. To construct the dataset, we randomly sample 20k data from the simulation results. The training set includes 90% of them, and the remaining data are used as the testing set. We use 8 linear layers to parameterize the derivative of the Neural ODE, with each layer containing 56 neurons. Mini-batch BN, Pop-TI BN, and TA-BN are applied to the first 3 layers. We use $M = 100$ for TA-BN. The final linear layer outside the Neural ODE maps the 56 features to 40 prediction values. For ODE solving, we use the dopri5 method with a tolerance level of $10^{-3}$. We use the SGD optimizer with a learning rate of $10^{-2}$ to train the model for 16 epochs. The batch size is 200.

Figure 9 presents the mean square error (MSE) results of Neural ODEs w/o BN, w/ Mini-batch BN, w/ Pop-TI BN, and w/ TA-BN. BN can improve the prediction, and TA-BN surpasses Mini-batch BN. It indicates that TA-BN is suitable for the circuit modeling task.

## 6   Conclusions and Limitations

In this paper, we demystify the previously unknown reason why batch normalization (BN) may lead to performance degradation when used with Neural ODEs. The fundamental mismatch arises from BN being initially designed for discrete neural networks without a temporal dimension, whereas Neural ODEs operate continuously over time. To address this challenge, we propose temporal adaptive batch normalization (TA-BN), specifically tailored for Neural ODEs. Our key is associating population statistics and learnable parameters with predefined regular time grids, with temporal interpolation automatically performed during training and testing. As a continuous-time counterpart to traditional BN, TA-BN ensures training stability, thereby facilitating the scaling of Neural ODE model sizes. Our extensive experiments demonstrate TA-BN's ability to enhance the performance of Neural ODEs compared to baseline methods in tasks such as image classification and physical system modeling.

The interpolation process used in TA-BN inevitably introduces runtime overhead, which slows down the execution of Neural ODEs. The parameter-free linear interpolation technique is empirically found to be stable, time-efficient, and capable of providing performance improvement. However, future work can focus on enhancing the temporal interpolation used in TA-BN to further optimize its efficiency and performance.

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

# A   Implementation Details

## A.1   Image Classification

Our model comprises a Neural ODE module followed by a single linear layer, which we refer to as the `unmixed` Neural ODE architecture. The single linear layer maps the activations provided by Neural ODE to the final class probability (or logit). We have 3 settings for the Neural ODE module.

**Multilayer Perceptron (MLP)**: We use 2 linear layers inside the Neural ODE module. On MNIST, each linear layer has $28 \times 28$ hidden features, matching the number of pixels in an MNIST image.

**Convolutional Neural Network (CNN)**: We use $4 \sim 12$ convolution layers inside the Neural ODE module. The kernel size of all layers is $3 \times 3$. The numbers of output channels are:

4 layers: $3(\text{input}) \to 32_\downarrow \to 64_\downarrow \to 32_\uparrow \to 3_\uparrow$.
6 layers: $3(\text{input}) \to 32_\downarrow \to 64_\downarrow \to 128_\downarrow \to 64_\uparrow \to 32_\uparrow \to 3_\uparrow$.
$7 + L_c$ layers: $3(\text{input}) \to 32_\downarrow \to 64_\downarrow \to 128_\downarrow \ (\to 256) \times L_c \to 128 \to 64_\uparrow \to 32_\uparrow \to 3_\uparrow$.

Note that a layer with $\downarrow$ downscales the feature maps using a stride of 2. A layer with $\uparrow$ is a transposed convolution layer that upscales the feature maps using a stride of 2.

**U-Net**: We use a U-Net with 18 convolution layers. The kernel size of all layers is $3 \times 3$. The numbers of output channels are designed as $3(\text{input}) \to 32_\downarrow \to 32 \to 64_\downarrow \to 64 \to 128_\downarrow \to 128 \to 256 \to 256 \to 128_\uparrow \to 128 \to 128 \to 64_\uparrow \to 64 \to 64 \to 32_\uparrow \to 32 \to 32 \to 3$.

## A.2   Physical Dynamical Systems Modeling

The HalfCheetah-v2 and Walker2d-v2 datasets consist of trajectories of 3D robot systems generated by the Mujoco physics engine [40]. Each trajectory represents a sequence of a 17-dimensional vector describing the system's state, such as the robot's joint angles and poses. The Neural ODEs are required to predict the trajectories in an autoregressive manner, as in [25, 42].

In these tasks, the input size is 17, and the Neural ODE module is designed as $17(\text{input}) \ (\to 64) \times 11 \to 17$. BN is applied to the first 5 layers. Based on the code from [42], we do the prediction via the invariance set propagation and data-controlled neural ODE proposed in [42]. Since the input size is the same as the output size, we don't need a linear layer outside the Neural ODE module.

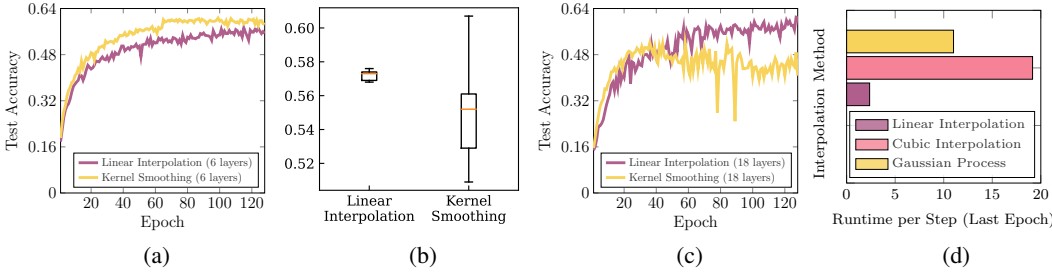

Figure 10: (a) Kernel smoothing can achieve slightly better accuracy on CIFAR-100 with 6 convolution layers for learnable derivatives. (b) Linear interpolation achieves more stable performance across 5 runs. (c) Linear interpolation performs better with a larger model size. (d) We compare different methods on the runtime per step at the last epoch to show the efficiency of linear interpolation.

# B   Discussion on Interpolation Methods

In Figure 7, we compare the performance of four interpolation methods, including linear interpolation, cubic interpolation, kernel smoothing, and Gaussian process. The cubic interpolation and kernel smoothing methods achieve good performance with 6 convolution layers for learnable derivatives. The results on CIFAR-100 also support this point, as shown in Figure 10(a). However, Figure 10(b) indicates that linear interpolation achieves more stable performance, and Figure 10(c) clearly shows the superiority of linear interpolation with a larger model size. Furthermore, linear interpolation is more efficient than cubic interpolation and Gaussian process, as shown in Figure 10(d). Therefore, we use linear interpolation by default, which can provide superior performance and stability.

# C  Additional Ablation Studies

## C.1  Ablation Study on ODE Solvers

We have conducted extra experiments on CIFAR-10 using fixed-step solvers like Euler method, with the 8-layer backbone. The results are shown in Table 5. Regardless of the solver, TA-BN achieves the best performance among the techniques. We also explored midpoint and rk4 solvers; however, they are much slower and haven't finished in the limited time constraint.

## C.2  Ablation Study on Time Grids

Regarding the grid size hyperparameter $M$, we ran experiments without BN and found that the number of function evaluations (NFE) is around hundreds. Thus, we set $M = 100$ in our paper. We have performed extra ablation studies on it, as reported in the following table. Using $M > 100$ brings no improvement but too much runtime overhead. We don't have the confidence interval of $M = 500$ due to the time limit.

Table 4: Ablation study on ODE solvers.

| Method | ODE Solver | Accuracy |
|---|---|---|
| w/o BN | Euler | 0.839±0.002 |
| w/ Pop-TI BN | Euler | 0.631±0.203 |
| w/ Mini-batch BN | Euler | 0.864±0.002 |
| w/ TA-BN (ours) | Euler | 0.872±0.003 |
| w/o BN | Dopri5 | 0.843±0.004 |
| w/ Pop-TI BN | Dopri5 | 0.332±0.090 |
| w/ Mini-batch BN | Dopri5 | 0.865±0.004 |
| w/ TA-BN (ours) | Dopri5 | 0.874±0.001 |

Table 5: Ablation study on ODE solvers.

| Method | Time Grids $M$ | Accuracy |
|---|---|---|
| w/ TA-BN (ours) | 10 | 0.851±0.015 |
| w/ TA-BN (ours) | 50 | 0.851±0.019 |
| w/ TA-BN (ours) | 100 | 0.874±0.001 |
| w/ TA-BN (ours) | 500 | 0.870 |

