# OpenReview forum: "Improving Neural ODE Training with Temporal Adaptive Batch Normalization"
_NeurIPS.cc/2024/Conference — NeurIPS 2024 poster_

### Official Review · Reviewer_XMSL · 2024-06-21

**Soundness:** 3
**Presentation:** 4
**Contribution:** 3
**Rating:** 7
**Confidence:** 4

**Summary:**

This paper focusses on adapting standard Batch Normalization to be applied to Neural ODEs. The paper proposes the reason for standard BN failing on Neural ODEs is that the population statistics cannot be meaningfully tracked for continuous $t$. That is, since Neural ODEs can be viewed as having continuous depth, the population statistics should also be time-dependent.

The proposed solution is to track the population statistics on a predetermined uniform time grid. During an ODE solve, for arbitrary $t$, interpolation on this grid is carried out to get $\mu(t)$ and $\sigma(t)$. This is also done for the learnable scale and locations of each Temporal BN layer. The interpolation is also used to update the population statistics on the grid based on batch statistics calculated at arbitrary times during training.

Evaluation on image classification and time-series regression tasks demonstrate that this adapted BN is effective and mitigates the issues with standard BN.

**Strengths:**

- The paper is nicely written.
- A meaningful problem has been identified, explored and a solution has been proposed. The solution is elegant in its simplicity.
- The evaluation is extensive and convincing.

**Weaknesses:**

There are very few weaknesses of this paper. My general opinion is that it should be accepted, ultimately the main weakness is that the work is "incremental". This is not a major weakness in my view it only limits the paper attaining the top scores.

In terms of weaknesses that can be addressed in a reasonable rebuttal period:
- As given in part 7 of the checklist, there are only error bars included for some of the experiments, but not all. Are the numbers available? More detail needs to be given about how many repeats are carried out, and if repeats have not been carried out for some experiments, this should be justified. This applies to all results given in the main paper.
- It would also be good to see how this method of BN can be applied to Neural CDEs and Continuous Normalizing Flows. This is not necessary but would still improve the paper.
- It seems like this method would not be necessary if we train and carry out inference with fixed solver step size. Since here the population statistics can be stored and are meaningful at these points. This will negatively impact the accuracy of the solve, however it would be good to see this tradeoff in another experiment if possible.

**Questions:**

- How have hyperparameters been selected?
- I'm interested to hear a deeper explanation about why standard BN fails. The dynamics function of the Neural ODE is $f_\theta(x, t)$. Assume this is an MLP with one hidden layer, and we only apply BN at the hidden layer. Aren't the batchnorm statistics more about tracking the mean and variance of the hidden state of this dynamics function, across the distribution of all possible hidden states given $x$ and $t$. And so, since this applies to the dynamics function, as long as many possible $x(t)$ and $t$ are seen during training BN is still valid? If so couldn't the training issues be avoided with lower learning rate or gradient clipping. My question is essentially: is this an engineering trick to improve training, or a solution to a deeper fundamental problem with applying BN to Neural ODEs? Either answer is fine, I'm just curious.

**Limitations:**

- The limitations are addressed in the conclusion.
- There is no broader impact statement, it is not necessary for this work but would still improve the paper.

---

> ### Author Rebuttal · Authors · 2024-08-07
>
> We appreciate the insightful feedback from Reviewer XMSL. Below we respond to each raised concern.
>
> ---
> **Q0.1: Error bar.**
>
> We had some error bar results in Fig. 10 in the Appendix. We omitted others in the main text, as we observed TA-BN's stability across independent runs. To justify it, we add the repeated results in the following table. We agree with the reviewer on the importance of error bars. In the revised manuscript, we will update **the whole Table I** using the format of 'mean±std'.
>
> |Method|Dataset|Model|Accuracy|
> |-|-|-|-|
> |TA-BN|CIFAR10|8-layer|0.874±0.001|
> |TA-BN|CIFAR10|UNet|0.910±0.010|
> |Mini-batch BN|CIFAR10|UNet|0.822±0.095|
> |Pop-TI BN|CIFAR10|UNet|0.548±0.087|
> |w/o BN|CIFAR10|UNet|0.517±0.049|
> |TA-BN|SVHN|UNet|0.958±0.004|
> |Mini-batch BN|SVHN|UNet|0.906±0.310|
> |Pop-TI BN|SVHN|UNet|0.241±0.123|
> |w/o BN|SVHN|UNet|0.096±0.025|
>
> ---
> **Q0.2: How TA-BN can be applied to Neural CDEs and CNFs?**
>
> Neural CDE is for time series, and Continuous Normalizing Flow (CNF) is for density matching and generation tasks. We considered integrating TA-BN with them initially but realized it was improper for three reasons: (i) BN is mostly used for image problems. Other normalizations can be applied (e.g., layer norm), but these are already distant from BN. As an extension of BN, TA-BN may not be suitable for time series and density matching. (ii) Neural CDE and CNF often use shallow models that perform adequately, and adding TA-BN does not yield significant improvement. (iii) The need for integrating TA-BN into high-dimensional CNF has been eliminated by flow matching [1], where CNF no longer needs ODE solving.
>
> ---
> **Q0.3: TA-BN is not necessary with a fixed step size.**
>
> We view TA-BN as being simplified in a fixed step-size solver rather than unnecessary. In such solvers, TA-BN degenerates to a simpler version without interpolation, aligning with the reviewer's point on storing statistics at fixed time grids. However, standard BN will still malfunction with a fixed step-size solver. For detailed explanations of each BN technique in the Neural ODE context, please refer to our response to Q2.
>
> We have tried different ODE solvers, including fixed step-size ones, and added the results in the table below. Regardless of the ODE solver, TA-BN is the best one compared to other techniques. We also tested midpoint and rk4 solvers, but they are much slower and didn’t finish within our time constraint.
>
> |Method|Model|ODE Solver|Accuracy|
> |-|-|-|-|
> |TA-BN|8-layer|dopri5|0.874±0.001|
> |Mini-batch BN|8-layer|dopri5|0.865±0.004|
> |Pop-TI BN|8-layer|dopri5|0.332±0.090|
> |w/o BN|8-layer|dopri5|0.843±0.004|
> |TA-BN|8-layer|euler|0.872±0.003|
> |Mini-batch BN|8-layer|euler|0.864±0.002|
> |Pop-TI BN|8-layer|euler|0.631±0.203|
> |w/o BN|8-layer|euler|0.839±0.002|
>
> ---
> **Q1: How to select hyperparameters?**
>
> Below we briefly show our hyperparameter setting criteria and extra ablation studies on hyperparameters. They will be added to the updated manuscript.
>
> + We chose the popular dopri5 ODE solver. Since the ODE tolerance impacts the accuracy and runtime, we tested tolerances in $[10^{-5},10^{-1}]$ and chose $10^{-3}$, which has decent accuracy and affordable solving time. Additional experiments on different solvers can be found in our response to Q0.3 above.
>
> + Regarding the number of TA-BN time grids $M$, we ran experiments without BN and found that the number of function evaluations (NFE) is usually around hundreds. Thus, we used $M=100$ in our manuscript. Ablation studies on the value of $M$ for TA-BN are shown in the following table. The confidence interval of $M=500$ is absent due to the time limit.
>
> |Model|Time Grids|Accuracy with TA-BN|
> |-|-|-|
> |8-layer|10|0.851±0.015|
> |8-layer|50|0.851±0.019|
> |8-layer|100|0.874±0.001|
> |8-layer|500|0.870|
>
> + In image classification, we used typical hyperparameters (e.g., AdamW optimizer with a learning rate of 1e-3). We train models for 128 epochs to ensure convergence. The learning rate is decreased by a factor of 0.1 at the 64th epoch.
>
> + In physical systems modeling, we empirically selected the number of MLP layers to make the models deep enough so that the effect of BN can be observed. We tried different solvers like AdamW, RMSprop, and SGD with typical learning rates and selected the best one.
>
> ---
> **Q2: Deeper explanation on why standard BN fails.**
>
> Here, we present clear pseudocode and deep illustrations. Considering the Neural ODE mentioned by the reviewer: $\\frac{dx}{dt}=f_\\theta(x,t)$, where $f_\\theta$ is an MLP with one hidden layer and standard BN, its `forward` function is:
>
> ```python
> def forward(x,t):
>     a1 = ReLU(self.linear1(x))
>     a2 = self.bn(a1)
>     return self.linear2(a2)
> ```
> When solving from the input $x(0)$ to the output $x(T)$, this model will be called $N$ times at different time points $\\lbrace t_n \\rbrace_{n=1}^N$. This forces `self.bn` to update its statistics based on all $\\lbrace t_n \\rbrace_{n=1}^N$. Essentially, it assumes $a_1$ has the same distribution regardless of $t$, which is not true. This is the problem of applying standard BN to Neural ODE in any DL frameworks, which we call **Pop-TI BN** in our manuscript. The issue is more profound than an engineering problem, and simple variants of BN also fail. First, **Mini-batch BN** uses $a_1$'s mini-batch statistics at each time step. It suffers from outliers and small batches. Second, **Pop BN** uses $N$ statistics for $\\lbrace t_n \\rbrace_{n=1}^N$ separately. It doesn't work because adaptive solvers change $\\lbrace t_n \\rbrace_{n=1}^N$ for every batch. For fixed-step-size solvers, Pop BN can work, and our TA-BN is equivalent to Pop BN in this case.
>
> ---
> **Q3: Broader impact statement.**
>
> We have reached our response length limit but will include a broader impact statement in the updated manuscript.
>
> ---
> **References:**
>
> [1] Yaron Lipman et al., 'Flow Matching for Generative Modeling', ICLR 2023.

---

> > ### Comment · Reviewer_XMSL · 2024-08-07
> > **Thank you for the response, my score remains the same**
> >
> > Thank you for the detailed response. I have now read all reviews and responses.
> >
> > I maintain my view that this is a strong paper and maintain my score.
> >
> > In particular I am grateful for the explanation of why we need Temporal Batchnorm rather than just normal batchnorm.
> >
> > However, I am not convinced by the argument that TBN may not be suitable for time-series and CNFs. SInce the exact same argument can be made that continuous depth models are not suitable for image recognition.

---

> > > ### Author Response · Authors · 2024-08-07
> > >
> > > Thank you for your reply. We will consider applying TA-BN to time series and generation in future work. Your suggestion is valuable and sincerely appreciated. In addition to the reasons provided in the reply, we would like to note that we did conduct quick, small-scale experiments on TA-BN with the mentioned models, and our preliminary results did not exhibit significant improvements.

---

### Official Review · Reviewer_97fY · 2024-07-11

**Soundness:** 2
**Presentation:** 3
**Contribution:** 2
**Rating:** 4
**Confidence:** 3

**Summary:**

The paper presents Temporal Adaptive Batch Normalization (TA-BN) which is tailored for Neural Ordinary Differential Equations (Neural ODEs). This method addresses the limitation of applying traditional Batch Normalization to Neural ODEs by acting as a continuous-time analog. The use of TA-BN in Neural ODEs is shown to allow for deeper architectures, which in turn improves performance. The paper demonstrates the effectiveness of TA-BN in image classification tasks, achieving a high test accuracy on CIFAR-10 comparable to more complex models, and also shows its advantage in physical system modeling.

**Strengths:**

- The writing and presentation of this paper are good.
- This paper studies the issue of using BN in Neural ODEs, which is an interesting topic.
- The experiment results are good.

**Weaknesses:**

- The comparison to related works in the context of SNN is not enough. Especially, I found TAB is highly related to this paper (even though the name is highly similar). There should be a comprehensive comparison to clarify the novelty and contribution of this paper.

- Lack of experiments on training efficiency.


[1] TAB: Temporal Accumulated Batch Normalization in Spiking Neural Networks;

**Questions:**

- What is the impact of TA-BN on the training efficiency? I observe this paper selected STEER as a baseline, however, STEER is designed to accelerate the Neural ODEs instead of enhancing its performance.

**Limitations:**

See weakness. I am willing to increase my score, if my concern can be well-addressed.

---

> ### Author Rebuttal · Authors · 2024-08-07
>
> We sincerely appreciate reviewer 97fY for dedicating time to review our paper. The thoughtful feedback and constructive comments have been invaluable in improving the quality of our work.
>
> ---
> **Q1: Comparison to related works in the context of SNN and clarifications on novelty.**
>
> Thank you for the constructive feedback. In the submitted manuscript, we cited several papers on BN applied in SNN, including the TAB paper mentioned by the reviewer. Both TAB and TA-BN design BN by considering the temporal characteristics of models, but they differ significantly in the following aspects. The proposed TA-BN employs a temporal adaptive scheme through interpolation and is specifically designed for Neural ODEs with any type of solver (i.e., fixed step-size or adaptive step-size). In contrast, TAB [1] is designed for an SNN using fixed step-size discretization on the LIF neuron model, accumulating statistics based on all time steps visited up to the current time point: $\\mu_{1:t} = \\frac{1}{t} \\sum_{s=1}^{t} \\mu[s]$ and $\\sigma^2_{1:t} = \\frac{1}{t} \\sum_{s=1}^{t} \\sigma^2[s]$. Namely, TAB only records $\\{\\mu[1],\\mu[2],\\cdots,\\mu[t]\\}$ on uniform time grids, and thus **TAB cannot be direcly applied** in the Neural ODE context when adaptive solver is used, since we will need values like $\\mu[1.3]$, which is not available. Moreover, the learnable parameters in the BN layers at different time points are independent in TAB ($\\gamma[t]$ and $\\beta[t]$ in their symbols), while our proposed TA-BN also applies the temporal interpolation for them ($\\gamma$ and $\\alpha$ in our symbols).
>
> Regarding our contributions, Reviewer 6xoP appreciates our "thorough analysis of why traditional BN fails in Neural ODEs" and our "well-motivated" methodology. Reviewer XMSL concurs that we address a "meaningful problem" with a solution that is "elegant in its simplicity." With all due respect, we would like to justify our work from the following three main perspectives:
>
> + Although it is natural to consider applying BN to Neural ODEs, all preliminary efforts unfortunately fail to obtain stable and superior results. We for the first time thoroughly demystify the underlying reasons.
>
> + We proposed temporal accumulated BN (TA-BN) to resolve the above problem. TA-BN's time-dependent statistics and interpolation method can accurately normalize Neural ODEs' continuous-time dynamics.
>
> + Most Neural ODE studies use *mixed* structure. For instance, they insert a Neural ODE module into a middle layer of a CNN for image classification. However, the specific contribution of the Neural ODE module in such a setting remains unclear. Alternatively, prior works using *unmixed* structure (a pure Neural ODE followed by a learnable linear projection), such as Augmented Neural ODE, show poor performance. We achieve scalable and performant *unmixed* structures for the first time.
>
> ---
> **Q2: Lack of experiments on training efficiency.**
>
> Thank you for the constructive feedback. To demonstrate the impact of TA-BN on training efficiency, we compare the convergence speed and accuracy in the following table. When compared to Mini-batch BN, TA-BN can converge faster (CIFAR10 and CIFAR100) or with significantly higher accuracy (SVHN). As shown in the paper (e.g. Figures 3 and 5), Pop-TI BN and w/o BN are usually unstable, and some of them cannot converge. Therefore, TA-BN exhibits superior training efficiency. Moreover, although TA-BN inevitably introduces a small computational overhead, the overall training time may not increase due to the faster convergence speed. To better address the reviewer's concern, we will add the above discussion in the updated manuscript.
>
> Table. Convergence (defined as no improvement in 10 epochs) comparison
>
> | Dataset  | Model   | Method        | Epoch | Accuracy |
> |--------|-------|-------------|-----|--------|
> | MNIST    | 8-layer | TA-BN         | 40    | 0.984    |
> | MNIST    | 8-layer | Mini-batch BN | 40    | 0.984    |
> | CIFAR10  | 8-layer | TA-BN         | 89    | 0.868    |
> | CIFAR10  | 8-layer | Mini-batch BN | 92    | 0.864    |
> | SVHN     | UNet    | TA-BN         | 47    | 0.956    |
> | SVHN     | UNet    | Mini-batch BN | 33    | 0.920    |
> | CIFAR100 | UNet    | TA-BN         | 88    | 0.588    |
> | CIFAR100 | UNet    | Mini-batch BN | 100   | 0.584    |
>
> ---
> **Q3: Is STEER a proper baseline?**
>
> Thank you for your constructive feedback on our experimental details. STEER [2] employs random sampling of the end time of the ODE during training, which is a regularization technique that, as the reviewer mentioned, can accelerate training. However, as also stated in their abstract, and here we quote, '...the proposed regularization can significantly decrease training time and **even improve performance over baseline models**.' Therefore, we respectfully point out that STEER is a proper baseline. From a high-level perspective, both STEER and our proposed TA-BN are techniques that can improve Neural ODE performance, and thus they should be compared.
>
> ---
> **Reference:**
>
> [1] H. Jiang et al. 'TAB: Temporal Accumulated Batch Normalization in Spiking Neural Networks,' ICLR 2024.
>
> [2] A. Ghosh et al. 'STEER: Simple temporal regularization for neural ode.' NeurIPS, 2020.

---

> > ### Comment · Reviewer_97fY · 2024-08-09
> >
> > Thank you for your response. I have read your response and other reviews. I think it is natural to extend TAB via interpolation techniques. This to some extent reduces the novelty and impact of this work. Moreover, for the training efficiency experiments, what I like to see is the comparison regarding NFE, since this is actually the key fact of the training efficiency for Nerual ODE. Hence, I would like to keep my score.

---

> > > ### Author Response · Authors · 2024-08-09
> > > **Thanks for your response and clarification on our contribution**
> > >
> > > Thank you for your thoughtful feedback.
> > >
> > > We acknowledge the reviewer's observation that our proposed TA-BN shares several similarities with TAB (as we detailed in our response to Q1). However, we would like to respectfully emphasize that TA-BN is not our sole contribution. Another significant contribution is our demonstration of **why standard BN fails in Neural ODEs. This failure explanation can not be inferred from the TAB work in SNNs**, and we kindly ask the reviewer to consider this key aspect.
> > >
> > > Regarding training efficiency, since no specific metric was mentioned in the rebuttal, we opted to use the converged epoch, as it is a commonly accepted standard in traditional deep learning research. We sincerely apologize for any confusion this may have caused, as it differs from the NFE metric the reviewer had in mind. We will conduct experiments reporting the NFE metric, although it may not be possible to meet the discussion deadline. We will promptly update the reviewer with the results once available.

---

### Official Review · Reviewer_EgcE · 2024-07-11

**Soundness:** 3
**Presentation:** 3
**Contribution:** 2
**Rating:** 6
**Confidence:** 4

**Summary:**

The paper proposes a remedy for batch normalisation in NeuralODE training, which proposes using a time grid for estimating the depth-dependent statistics, which leads to improved accuracy of the model.

**Strengths:**

Clarity: the paper is clearly written, with good motivation

Quality: the quality of the paper is good in general, however, see the weaknesses section

Significance: training NeuralODEs in a more efficient way, which is also practical and computationally efficient, would be a significant insight for the community. However, this idea lacks back up for the given insights: please refer to the originality section of the weaknesses for more discussion.

**Weaknesses:**

Originality: the authors need to confirm the originality of the paper: it seems like as the idea of using time grid for batch normalisation parameters is intuitive, there needs to be more empirical/theoretical insight drawn from this observation to make it a full NeurIPS paper. Now, such insight seems to be limited and boils down to the statement that grid-wise batch normalisation parameters improve when they are selected using grid. How would such model behave in case of different ODE solvers, would it influence the efficacy of batch normalisation? There is also a need to explore the relationship of such model with the ResNet with coupled layer parameters, which is an Euler discretisation of the NeuralODE (see Chen et al, 2018) : does the model still give advantages over such model? Does the same effect on batch normalisation  repeat in such scenario?

Quality: the experimental results seems to not show any confidence intervals. They also do not seem to provide the answer on how to select the grid size hyperparameter (although I may have misunderstood it, so the author's clarification would be much appreciated).

**Questions:**

1) Piece-wise linear interpolation of the batch parameters, as outlined in Algorithm 1, would not be differentiable at the grid points (see line 2 of the algorithm); does it cause any problems for the convergence at the ODE integration?

2) The authors claim that '“These models exhibit several intriguing features, such as the ability to compute gradients with constant memory consumption”

I don’t think it would be an intriguing feature of Neural ODEs as  it has been achieved by coupling of NeuralODE parameters throughout the depths, which also could be done with the standard ResNets as well. Furthermore, the authors themselves define a time-dependent batch normalisation parameters grid which essentially highlights the trade-offs between the number of parameters and memory consumption vs accuracy.

3) There has been a line of work on how to parametrise NeuralODEs in a way that allows for continual change of parameters, which includes, e.g. shooting methods (Kwitt et al, 2020) and hypernetworks (the original paper Chen et al, 2018 itself, page 5, last paragraph of the intro in the section 4 called Time-dependent dynamics). I guess such line of work should be referenced as well, perhaps even showing whether such an alternative parameterisation through shooting methods/hypernetworks can equally address the problem.

Kwitt et al, A Shooting Formulation of Deep Learning, NeurIPS 2020

**Limitations:**

More empirical analysis would be necessary to highlight the failure modes: is such observed improvement of efficacy of batch normalisation observed for different ODE solvers? How does it compare with the hyper networks approach in the original NeuralODE paper?

---

> ### Author Rebuttal · Authors · 2024-08-07
>
> We extend our sincere gratitude to Reviewer EgcE for the constructive feedback. We will include the following discussions in the updated manuscript.
>
> ---
> **Q0.1 Originality: more insights and TA-BN with other ODE solvers**
>
> We have conducted extra experiments using fixed-step solvers like the Euler method, with the 8-layer backbone introduced in Appendix A.1. The results are shown in the following table. Regardless of the solver, TA-BN achieves the best performance compared to baselines. We also explored midpoint and rk4 solvers; however, they were too slow to finish within the limited time.
>
> |Method|Model|ODE Solver|Accuracy|
> |-|-|-|-|
> |TA-BN|8-layer|dopri5|0.874±0.001|
> |Mini-batch BN|8-layer|dopri5|0.865±0.004|
> |Pop-TI BN|8-layer|dopri5|0.332±0.090|
> |w/o BN|8-layer|dopri5|0.843±0.004|
> |TA-BN|8-layer|euler|0.872±0.003|
> |Mini-batch BN|8-layer|euler|0.864±0.002|
> |Pop-TI BN|8-layer|euler|0.631±0.203|
> |w/o BN|8-layer|euler|0.839±0.002|
>
>
> Regarding our originality, with all due respect, we would like to justify our work from the following perspectives:
>
> + Although it is natural to consider applying BN to Neural ODEs, all preliminary efforts fail to obtain stable and superior results. We for the first time thoroughly demystify the underlying reasons.
>
> + We proposed TA-BN to resolve the above problem. Its time-dependent statistics and interpolation method can accurately normalize Neural ODEs' continuous-time dynamics.
>
> + Although prior works using *unmixed* structure (a Neural ODE module followed by a linear layer) enable the clear analysis of Neural ODE architectures, they show poor performance. We achieve scalable and performant *unmixed* structures for the first time.
>
> ---
> **Q0.2 Quality: confidence interval and grid size hyperparameter**
>
> In our original manuscript, we included a set of results on confidence intervals (e.g., Fig. 10(b)) in the Appendix. We omitted others in the main text, as we empirically observed our method to be stable across independent runs. To justify it, we have added the repeated results in the following table, with the backbones 8-layer and UNet introduced in Appendix A.1. According to the results, TA-BN consistently outperforms other methods in accuracy and stability. We will update **the whole Table I** in our revised manuscript using the format of 'mean±std'.
>
> |Method|Dataset|Model|Accuracy|
> |-|-|-|-|
> |TA-BN|CIFAR10|8-layer|0.874±0.001|
> |TA-BN|CIFAR10|UNet|0.910±0.010|
> |Mini-batch BN|CIFAR10|UNet|0.822±0.095|
> |Pop-TI BN|CIFAR10|UNet|0.548±0.087|
> |w/o BN|CIFAR10|UNet|0.517±0.049|
> |TA-BN|SVHN|UNet|0.958±0.004|
> |Mini-batch BN|SVHN|UNet|0.906±0.310|
> |Pop-TI BN|SVHN|UNet|0.241±0.123|
> |w/o BN|SVHN|UNet|0.096±0.025|
>
> Regarding the grid size hyperparameter $M$, we ran experiments without BN and found that the number of function evaluations (NFE) is around hundreds. Thus, we set $M=100$ in our paper. We have performed extra ablation studies on it, as reported in the following table. Using $M>100$ brings no improvement but too much runtime overhead. We don't have the confidence interval of $M=500$ due to the time limit.
>
> |Method|Model|Time Grids|Accuracy|
> |-|-|-|-|
> |TA-BN|8-layer|10|0.851±0.015|
> |TA-BN|8-layer|50|0.851±0.019|
> |TA-BN|8-layer|100|0.874±0.001|
> |TA-BN|8-layer|500|0.870|
>
> ---
> **Q1: Non-differentiability of piecewise linear interpolation at grid points**
>
> If a variable $a=a(t)$ is defined piecewise linearly on time, then $\\frac{da}{dt}$ at defining time grids will be non-differentiable. However, our algorithm's backward propagation during training **does not involve such time-based gradients.** Specifically, using the symbols in Algorithm 1 of our manuscript, the time variables occur only in $\\omega_1$ and $\\omega_2$. According to the chain rule, we have $\\frac{dL}{d\\gamma_l^\\star}=\\frac{dL}{d\\gamma_j}\\frac{d\\gamma_j}{d\\gamma_l^\\star}=w_1\\frac{dL}{d\\gamma_j}$ for line 5, which doesn't involve any gradients with respect to time, and thus fully differentiable.
>
> ---
> **Q2: Constant memory consumption claimed by Neural ODE**
>
> We would like to clarify that the statement mentioned by the reviewer was intended as an introduction to Neural ODEs. This statement is **not related to**, nor a new property from our TA-BN, but is **directly referenced from** the original Neural ODE paper [1]. In that context, constant memory consumption means no intermediate results need to be stored during training because gradients are calculated by solving ODE (i.e., adjoint method) instead of backward propagation. We agree with the reviewer that this statement is confusing and will remove it in the updated manuscript.
>
> ---
> **Q3: Comparison with hypernetwork and shooting methods**
>
> Thanks for highlighting these methods. The shooting method [2] provides a time-varying weight trajectory. Hypernetwork [1] uses time-dependent model weights $\\theta(t)$. Both methods make model parameters dependent on time, enhancing flexibility. Alternatively, TA-BN addresses a different aspect, using time-dependent statistics to normalize the time-dependent outputs of each layer, stabilizing training and enabling deeper architectures. Note that while [1] and [2] can improve flexibility, TA-BN is still necessary to address training instabilities in deep Neural ODEs. In essence, the shooting method and hypernetwork (i.e., time-varying model parameters) address a different issue from TA-BN (i.e., time-dependent statistics).
>
> ---
> **Q4: Limitations: More empirical analysis on TA-BN with other ODE solvers, and compare it with hypernetworks**
>
> We kindly refer the reviewer to our response to your Q0.1 for the question of TA-BN with other ODE solvers, and our response to your Q3 for the comparison with hypernetworks.
>
> ---
> **References:**
>
> [1] Qicky T.Q. Chen et al., 'Neural Ordinary Differential Equations', NeurIPS 2018.
>
> [2] Kwitt et al, 'A Shooting Formulation of Deep Learning', NeurIPS 2020.

---

> > ### Comment · Reviewer_EgcE · 2024-08-09
> >
> > Many thanks for the insightful rebuttal! I am going through all the responses to all the authors and will follow up with the message after.
> >
> > I have one follow-up question though.
> >
> > "In essence, the shooting method and hypernetwork (i.e., time-varying model parameters) address a different issue from TA-BN (i.e., time-dependent statistics)."
> >
> > My question was mainly: the proposed method uses grid to mitigate the issues of time-dependent statistics. Although shooting methods/hypernetworks address a different problem, which is time-varying model parameters, there is a possibility that addressing the time-varying model parameters problem as per, e.g., [1-2] may (or may not) help enable using the batch normalisation techniques and equally solve the problem of time-dependent statistics. It would be good if the authors clarify upon it.

---

> > > ### Author Response · Authors · 2024-08-09
> > >
> > > Thanks for acknowledging our response.
> > >
> > > To the best of our knowledge, the hypernet and shooting methods are different from TA-BN and they cannot solve the problem TA-BN addresses. TA-BN normalizes the output from the previous layer using time-dependent statistics at each time grid, resulting in a final distribution with a mean of zero and a standard deviation of one. By maintaining consistent distributions across layers, it can mitigate the issue of internal covariate shift, where significant changes in distributions can make training unstable. However, hypernet and shooting methods do not explicitly normalize the output distributions. Consequently, models using hypernet or shooting methods still suffer from internal covariate shift and require TA-BN for normalization.
> > >
> > > We would be grateful if the reviewer could kindly elaborate on how they believe the hypernet and shooting methods could be applied in our case. This would enable us to respond more effectively and precisely. We are open to further discussion.
> > >
> > > In the meantime, we hope that our other responses have addressed your concerns and that they lead to a more positive assessment.

---

> ### Comment · Reviewer_EgcE · 2024-08-09
>
> Many thanks for a swift response! Checking the other concerns in the meantime, and I'll update my score accordingly as soon as I finish (no questions on them yet, just need to read it carefully). Many thanks for preparing  a really thorough rebuttal.
>
> Just to elaborate on the question about TA-BN vs hypernets: I  understand that the hypernets and shooting methods are different from TA-BN. There is no batch normalisation in hypernet-parameterised neural ODEs,  it is instead a way to parameterise layers dynamically. My question is different: fundamentally, batch normalisation does not work with standard neural ODE with coupled layers, as the authors show. But if the problem with NeuralODEs and difference with resnets were solely in coupling the layers (which may or may not be true), then other methods which decouple the layers such as hypernets and shooting methods may potentially be used jointly with batch normalisation similarly to the proposed approach and deliver the same effect. Therefore, my question is whether the authors have any idea how does TA-BN compare over hypernets+BN or shooting+BN.

---

> > ### Author Response · Authors · 2024-08-09
> > **Thanks for the clarification and our further thoughts**
> >
> > Thanks so much for the swift response and the clarification! We really appreciate your time involved in the discussion. Now we understand the question better.
> >
> >
> > To summarize, hypernet and shooting methods (referred to as HS below) will replace the parameter $\theta$ with a time-varying parameter $\theta(t)$ in the Neural ODE equation, yielding $\frac{dx}{dt}=f(x(t),\theta(t))$, This approach decouples the layers and learnable parameters along the time axis.
> >
> >
> > Traditional BN involves not only learnable parameters but also the crucial aspect of obtaining batch population statistics during training to reuse during inference. However, HS methods, in their original formulation, do not define how to acquire these batch population statistics because they do not consider BN.
> >
> >
> > Going one step further, if we extend HS to make the population statistics of BN also time-dependent $\mu(t)$ and $\sigma(t)$ in $[0,T]$,  then it will encounter the same problem as POP BN we discussed in our manuscript, since not every $t$ in $[0,T]$ will be visited by the adaptive ODE solver during training, and at inference time, the required $\mu$ and $\sigma$ at $\\{t_1,t_2,…,t_N\\}$ might not be available. Thus, a temporal interpolation technique is inevitable for recording the population statistics $\mu(t)$ and $\sigma(t)$ correctly, which is exactly the TA-BN approach.
> >
> > The above discussion reflects our understanding on this question, and we hope it addresses the concern raised. We are keen to hear the reviewer’s further thoughts and insights on this matter, as your perspective is invaluable to us. We remain open to continued discussions to refine our explanations or clarify any remaining questions.

---

> > > ### Comment · Reviewer_EgcE · 2024-08-09
> > >
> > > Many thanks, it answers my question. I think it would be actually good to add this discussion, for example, into the appendix.
> > >
> > > Meanwhile, I'm looking at the rest of your rebuttal and will revise the score after.

---

> ### Comment · Reviewer_EgcE · 2024-08-11
>
> After reading the discussion with all the reviewers, I would like to thank the authors for addressing the outstanding concerns. New results look reasonable, confirm the hypothesis, and improve the value of the paper.
>
> I think the paper has scope and value of both presenting the reasons behind the failure of BN in Neural ODEs and the remedy for it and deserves to be accepted.

---

> > ### Author Response · Authors · 2024-08-11
> >
> > Thank you very much for taking the time to review our work. Your insightful feedback and constructive suggestions are invaluable and greatly appreciated. Your support and encouragement mean a lot to us. We will incorporate our discussions into the updated manuscript.

---

### Official Review · Reviewer_6xoP · 2024-07-12

**Soundness:** 3
**Presentation:** 3
**Contribution:** 3
**Rating:** 6
**Confidence:** 4

**Summary:**

This work identified the fundamental mismatch between Neural ODEs and traditional batch normalization (BN) techniques. To address this issue, the authors introduced Temporal Adaptive Batch Normalization (TA-BN), incorporating temporal interpolation to accumulate mini-batch statistics during training and use them as population statistics during inference. Neural ODEs with TA-BN outperformed those with traditional BN techniques and without BN on image classification and physical system modeling tasks. Additionally, TA-BN enhanced the performance of some other Neural ODE variants.

**Strengths:**

The work provided a thorough analysis of why traditional BN fails in Neural ODEs, which adds depth to the understanding of the problem.

The proposed TA-BN is well-motivated and has the potential to improve the performance of Neural ODEs in multiple fields.

This paper is well-organized and well-written.

**Weaknesses:**

- **Overclaim of experimental results**. The work claims that Neural ODEs can approach MobileNetV2-level efficiency in the abstract and introduction. However, training Neural ODEs can be slow, and TA-BN may exacerbate this problem due to increased computational complexity. Even if Neural ODEs with TA-BN achieve similar parameter efficiency to MobileNetV2, time efficiency should also be considered.

- **Lack of theoretical analysis**. The work focused heavily on empirical results but could benefit from a more in-depth theoretical analysis of why TA-BN works so well, potentially providing insights that could lead to further improvements.

- **Inappropriate experiment design**.  Vanilla Neural ODEs should not be used for Walker2d and HalfCheetah tasks due to potential collisions in such robotic systems [1]. Dynamical systems such as predator-prey equations and double pendulums are recommended.

- More ablation studies should be performed, such as testing Neural ODEs with traditional BN, with TA-BN, and without BN using fixed-step-size ODE solvers to support the claim that adaptive step-size ODE solvers cause the failure of traditional BN in Neural ODEs. Additionally, more NDE models with TA-BN, such as ODE-RNNs [2], Latent ODEs [2], and Neural CDEs [3], should be tested to demonstrate TA-BN's performance.

**References**:

[1] Chen, Ricky TQ, Brandon Amos, and Maximilian Nickel. "Learning Neural Event Functions for Ordinary Differential Equations." International Conference on Learning Representations.

[2] Rubanova, Yulia, Ricky TQ Chen, and David K. Duvenaud. "Latent ordinary differential equations for irregularly-sampled time series." *Advances in neural information processing systems* 32 (2019).

[3] Kidger, Patrick, et al. "Neural controlled differential equations for irregular time series." Advances in Neural Information Processing Systems 33 (2020): 6696-6707.

**Questions:**

Please see the comments above

**Limitations:**

Yes, it has discussed the limitations.

---

> ### Author Rebuttal · Authors · 2024-08-07
>
> We appreciate Reviewer 6xoP's thorough review and valuable insights. Below we respond to each raised concern.
>
> ---
> **Q1: Overclaim of experimental results about "MobileNetV2-level efficiency"**
>
> We will clarify in our updated manuscript that "efficiency" refers to the accuracy a model can achieve given the number of learnable parameters, i.e., **parameter efficiency**—a concept commonly used in tiny/efficient ML [1], [2]—**not training efficiency** mentioned by the reviewer. Our TA-BN can elevate an unmixed Neural ODE to MobileNetV2-level parameter efficiency. Alternatively, we agree with the reviewer that Neural ODEs, w/ or w/o TA-BN, is relatively slow due to ODE solving.
>
> ---
> **Q2: Lack of theoretical analysis on why TA-BN works.**
>
> We respectfully highlight that our explanation (e.g., Figure 2 and right of Figure 3) sufficiently reveals the inadequacy of traditional BN for Neural ODE, and why TA-BN works. We agree that a theoretical analysis would make our argument more convincing, which is provided below:
>
> The ideal normalization has mean $\\mu_k$ and variance $\\sigma^2_k$ at every time step $t_k$. Regardless of time, traditional BN uses the mean $\\bar{\\mu}$ and variance $\\bar{\\sigma}$ obtained by averaging the statistics over all time steps. We can express accurate ODE solving (e.g. Euler method) as:
> $$
> \\small
> h(t_n) = b \\sum_{k=0}^{n-1} ( \\frac{f_{\\theta} (h(t_k), t_k) - \\mu_k}{\\sigma_k} ), \\; i \\in [1, 2, \\cdots, N], \\; t_0 = 0, t_N = T.
> $$
> where $b$ is the time step, the input and output are $h(0)$ and $h(T)$, respectively. In traditional BN, we use a similar equation but replace $h(t_k)$, $\\mu_k$, and $\\sigma_k$ with $h_e(t_k)$, $\\bar{\\mu}$, and $\\bar{\\sigma}$, respectively.
>
> The difference between ideal normalization and traditional BN is $\\Delta_n = h(t_n) - h_e(t_n) = b \\sum_{i=0}^{n-1} \\delta_k$, where
> $
> \\delta_k = ( \\frac{f_{\\theta} (h(t_k), t_k)}{\\sigma_k} - \\frac{f_{\\theta} (h_e(t_k), t_k)}{\\bar{\\sigma}} ) - ( \\frac{\\mu_k}{\\sigma_k} - \\frac{\\bar{\\mu}}{\\bar{\\sigma}} ).
> $
> With some assumptions on $\\mu_k$, $\\sigma_k$ and $f_{\\theta}$, we can roughly prove $|\\Delta_n|$ has a lower bound grows asymptotically as $\\Omega (n)$.
>
> The above implies that traditional BN fails because time-independent statistics assumption will lead to increasing error. However, TA-BN does not suffer from it due to the time-dependent statistics, and the linear interpolation in TA-BN brings little errors to statistic estimation. Considering the interpolation of $\\mu$ as an example, to estimate mean $\\mu^{(t)}$ at time $t$, we use $G(t,\\mathbf{\\mu},\\mathcal{T})$ (Eq. (7) in our paper). Assuming that $\\mu$ is Lipschitz-continuous in the range $[t_l, t_{l+1}]$ with a Lipschitz constant $k_l$, we can derive the upper bound of error:
> $$
> \\Vert G(t,\\mathbf{\\mu},\\mathcal{T}) - \\mu^{(t)} \\Vert = \\Vert \\frac{t_{l+1}-t}{t_{l+1}-t_l} (\\mu_l - \\mu^{(t)}) + \\frac{t-t_l}{t_{l+1}-t_l} (\\mu_{l+1} - \\mu^{(t)}) \\Vert \\leq \\frac{t_{l+1}-t}{t_{l+1}-t_l} k_l (t - t_l) + \\frac{t-t_l}{t_{l+1}-t_l} k_l (t_{l+1} - t) \\leq \\frac{k_l}{2} (t_{l+1} - t_l).
> $$
> We can make the time interval $[t_l, t_{l+1}]$ small enough to have a small slope of $\\mu$, and thus a small $k_l$.
>
> Due to the time limit of response, we will leave a comprehensive detailed proof for future work.
>
> ---
> **Q3: Inappropriate experiment design. Vanilla Neural ODEs are not for Walker2d and HalfCheetah.**
>
> First, we would like to clarify that the Neural ODEs for Walker2d and HalfCheetah (Table 3), are based on Reference [42] in our paper. They **incorporate the special treatments proposed by [42] to avoid collisions, rather than employing vanilla Neural ODEs.** Concretely, [42] prevents the system from entering unsafe regions by enforcing constraints via invariance propagation. Based on [42], we employ a larger Neural ODE and test various BN techniques. We appreciate the reviewer's insight regarding predator-prey and double pendulums, which are valuable and will be considered in future research.
>
> ---
> **Q4: Ablation studies on fixed-step size solver and other Neural ODE models.**
>
> We have conducted extra experiments using fixed-step solvers like Euler method, with the 8-layer backbone introduced in Appendix A.1. The results are shown in following table. Regardless of the solver, TA-BN achieves the best performance among the techniques. We also explored midpoint and rk4 solvers; however, they are much slower and haven't finished in the limited time constraint.
>
> |Method|Model|ODE Solver|Accuracy|
> |-|-|-|-|
> |TA-BN|8-layer|dopri5|0.874±0.001|
> |Mini-batch BN|8-layer|dopri5|0.865±0.004|
> |Pop-TI BN|8-layer|dopri5|0.332±0.090|
> |w/o BN|8-layer|dopri5|0.843±0.004|
> |TA-BN|8-layer|euler|0.872±0.003|
> |Mini-batch BN|8-layer|euler|0.864±0.002|
> |Pop-TI BN|8-layer|euler|0.631±0.203|
> |w/o BN|8-layer|euler|0.839±0.002|
>
> We have considered using TA-BN with ODE-RNNs, Latent ODEs, and Neural CDEs. However, we realized it was not appropriate or out of our context because (i) These models are for irregular time series, while BN is mostly used for image problems. Variants of normalizations can be applied to time series (e.g., layer norm), but these are already distant from BN. As an extension of BN, TA-BN may not be suitable for time series. (ii) Neural ODE applications in time series often use shallow models with a small number of parameters (e.g., see Supplementary 5 of [3]). These models perform adequately in related tasks, and TA-BN does not bring significant improvement.
>
> ---
> **References:**
>
> [1] S. Han et al., 'Deep compression: Compressing deep neural networks with pruning, trained quantization and huffman coding,' ICLR 2016.
>
> [2] H. Mostafa et al., 'Parameter efficient training of deep convolutional neural networks by dynamic sparse reparameterization,' ICML 2019.
>
> [3] Y. Rubanova et al., 'Latent ordinary differential equations for irregularly-sampled time series,' NeurIPS 2019.

---

> > ### Comment · Reviewer_6xoP · 2024-08-07
> > **Thanks for your response-Raise the score**
> >
> > Thank you very much for your response. You have addressed most of my concerns, so I raise the score to 6.

---

> > > ### Author Response · Authors · 2024-08-08
> > >
> > > We sincerely appreciate your reply. Your review is immensely valuable and deeply appreciated.

---

### Author Rebuttal · Authors · 2024-08-07

We sincerely thank all the reviewers for their thoughtful and constructive feedback on our manuscript. We are encouraged by their positive remarks, noting that our proposed TA-BN is well-motivated (Reviewer 6xoP, Reviewer EgcE), addresses an interesting and important problem (all reviewers), and that our experiments are robust and convincing (Reviewer 97fY, Reviewer XMSL). We also appreciate the comments highlighting that the paper is well-organized and well-written (all reviewers). We diligently addressed all the concerns raised by providing ample evidence and the requested results. The raised points will be thoughtfully considered and integrated into the revised manuscript. Here is the summary of our responses to the raised questions:

1. **Extra numerical results (e.g., confidence interval, other ODE solvers)**: We have conducted additional experiments to analyze the confidence intervals of various methods, the effects of different ODE solvers, and the results of hyperparameter ablation studies, among others. Our TA-BN exhibits better accuracy and stability under different ODE solvers compared to baselines. The selected number of time grids makes a superior trade-off between accuracy and speed.

2. **TA-BN with other Neural ODE models (e.g., ODE-RNN, Neural CDE, CNF)**: Regarding the application of TA-BN with methods like ODE-RNNs [1], Neural CDEs [2], and continuous normalizing flow (CNF) [3], we did perform a quick, small-scale experiment. However, we realized it was not appropriate or out of our context due to the following reasons: (i) BN was originally proposed and is mostly used for image problems. As an extension of BN, TA-BN may not be suitable for time series or density matching. (ii) ODE-RNN, Neural CDE, and CNF often use shallow models with a small number of parameters as the trainable dynamics. These shallow models perform adequately well in related tasks, and adding TA-BN to these shallow neural networks does not yield significant improvement.

3. **Novelty and contribution:** Our TA-BN contributes to the field of Neural ODEs in the following three aspects: (1) We for the first time thoroughly demystify the underlying reasons why preliminary efforts of applying BN to Neural ODEs failed to obtain stable and superior results. (2) TA-BN's time-dependent statistics and interpolation method can accurately normalize Neural ODEs' continuous-time dynamics, resolving the above problem. (3) Although prior works using *unmixed* structure (a Neural ODE module followed by a linear layer) enable the clear analysis of Neural ODE architectures, they show poor performance. We achieve scalable and performant *unmixed* structures for the first time.

4. **Experimental setup (e.g., hyperparameters, baseline choices)**: We select hyperparameters based on prior research and our empirical experience. For physical system modeling baselines, we meticulously choose a base method that satisfies the system constraints before modifying the architecture and integrating different BN methods. These efforts ensure the effectiveness and validity of our experiments. Please refer to individual responses for details.

5. **Theoretical analysis**: We prove that traditional BN fails to normalize the activations with appropriate statistics, while TA-BN avoids this problem by accurately estimating the statistics at each time step. Furthermore, we prove that TA-BN's linear interpolation incurs negligible estimation error in statistics.

6. **Clarifications (e.g., efficiency, differentiability)**: To avoid misunderstanding, we clarify that "efficiency" in our discussion refers to the parameter efficiency in efficient/tiny-ML. For certain non-differentiable operations in TA-BN, we emphasize that training does not involve their gradients. We have also addressed other potential points of confusion concerning our baselines, Neural ODEs' memory consumption, and so on.

We would again like to thank all reviewers for their valuable time and feedback, and we hope that our changes adequately address all concerns. Any further questions are highly welcomed. Below, we will provide individual responses to address each reviewer’s concerns.


**References:**

[1] Y. Rubanova et al., 'Latent ordinary differential equations for irregularly-sampled time series,' NeurIPS 2019.

[2] P. Kidger et al. 'Neural controlled differential equations for irregular time series,' NeurIPS 2019.

[3] Qicky T.Q. Chen et al., 'Neural Ordinary Differential Equations', NeurIPS 2018.

---

### Decision · Program_Chairs · 2024-09-25

**Decision:**

Accept (poster)

**Comment:**

The reviewers all agreed on the premise of the paper: an algorithm for tracking Batch Normalization within the temporal field of Neural ODEs is introduced that uses interpolation. The paper explains the problems with naive implementation to BN and provides experiments comparing against other Neural ODEs benchmarks.

The reviewer scores average to a weak accept, but reviewer 97fY gave a borderline reject recommendation. This reviewer had a lower confidence score than the others. The reviewer opined that the work was not very novel compared to another published method (TAB) but the authors highlighted an additional insight from their work that did not appear in the TAB paper.
The existence of similar algorithms was also addressed by reviewer EgcE; the AC is also aware of similar algorithms for BN in Neural ODEs; reviewer XMSL also thought the paper could be considered incremental. These reviewers did not think that novelty / incrementality were blockers for publication.
 Additionally, after the discussion, reviewer 97fY insisted that additional comparisons on training efficiency would be necessary, but the authors did not think it would be possible to do so in the time allotted.

New experimental results were added during the discussion. These results were required for two of the reviewers to increase their score to 6. Therefore, it is necessary for the authors to include these results in their revision. The new experimental results are three distinct tables, so it should be possible to fit the results and a short discussion into the revision.

The authors also stated that a new proof in discussion would not be possible to include within time constraints. Including the proof was not required by the reviewer (6xoP) to increase the score.

Overall, the arguments to accept the paper outweigh the limitations noted by the reviewers. As such, the paper can be accepted (with the note that the reviewers did require the additional tables described above).